# Cargo Sorting at the *trans*-Golgi Network for Shunting into Specific Transport Routes: Role of Arf Small G Proteins and Adaptor Complexes

**DOI:** 10.3390/cells8060531

**Published:** 2019-06-03

**Authors:** Jing Zhi Anson Tan, Paul Anthony Gleeson

**Affiliations:** Department of Biochemistry and Molecular Biology and Bio21 Molecular Science and Biotechnology Institute, The University of Melbourne, Melbourne, VIC 3010, Australia; j.tan49@student.unimelb.edu.au

**Keywords:** *trans*-Golgi network, adaptor proteins, AP-1, AP-3, AP-4, GGAs, protein sorting, post-Golgi transport

## Abstract

The *trans*-Golgi network (TGN) is responsible for selectively recruiting newly synthesized cargo into transport carriers for delivery to their appropriate destination. In addition, the TGN is responsible for receiving and recycling cargo from endosomes. The membrane organization of the TGN facilitates the sorting of cargoes into distinct populations of transport vesicles. There have been significant advances in defining the molecular mechanism involved in the recognition of membrane cargoes for recruitment into different populations of transport carriers. This machinery includes cargo adaptors of the adaptor protein (AP) complex family, and monomeric Golgi-localized γ ear-containing Arf-binding protein (GGA) family, small G proteins, coat proteins, as well as accessory factors to promote budding and fission of transport vesicles. Here, we review this literature with a particular focus on the transport pathway(s) mediated by the individual cargo adaptors and the cargo motifs recognized by these adaptors. Defects in these cargo adaptors lead to a wide variety of diseases.

## 1. Introduction

In mammalian cells, the trans-Golgi network (TGN) has a distinct morphology, compared to the earlier Golgi cisternae, in that the TGN comprises networks of tubular, branching, and reticulating membranes [1,2]. The TGN has been typically viewed as a major sorting hub where cargoes are sorted into distinct transport carriers for trafficking to post-Golgi compartments and the plasma membrane (PM) [3,4] (Figure 1A). Moreover, the TGN also receives transport carriers from the endocytic trafficking pathway [5,6,7] (Figure 1A). Therefore, the TGN is a central organelle that intersects with both the biosynthetic and endocytic pathways. Given the multiple trafficking routes, transport machinery is employed to maintain the integrity of cargo sorting into the transport carriers. In addition, there is evidence that the cargoes may be segregated into specific TGN membrane micro-domains, and these microdomains are governed by small G proteins and TGN golgins [8,9,10]. Transport machinery, including small G proteins of the ADP ribosylation factor (Arf) family, cytosolic cargo adaptor proteins, coat proteins, and accessory proteins are key players in regulating cargo sorting at the TGN (reviewed in [3,4,6]) (Figure 1 and Figure 2). Central to the sorting process is the interaction of cytosolic adaptor proteins with specific motifs on the cytoplasmic domains of membrane cargo proteins. These adaptor proteins are recruited to the TGN membranes by active membrane-associated Arf small G proteins, which then allow them to bind cargoes (Figure 2). In some cases, soluble cargoes indirectly associate with adaptor proteins through specific transmembrane cargo receptors [3,4,6]. Sorting signals, such as the classical tyrosine-based YXXΦ (Φ denotes a bulky hydrophobic amino acid and X denotes any amino acid) and dileucine-based (DE)XXX(LL) signals, located at the cytoplasmic tail of cargoes or cargo receptors are important for the recognition and binding by adaptor proteins [11,12] (Figure 2). This interaction then allows for selective inclusion of cargoes into their respective transport carriers [11,12,13] (Figure 2B). Some adaptor proteins recruit coat proteins, such as clathrin, which then polymerize to form electron-dense coated membrane vesicles [14]. Adaptor proteins also recruit accessory proteins to drive membrane curvature and scission, which leads to vesicle formation [15,16,17,18,19,20]. Upon the release of transport vesicles from the membranes, coat proteins are dissociated and recycled for additional rounds of vesicle formation.

## 2. Small G Proteins of the ADP Ribosylation Factor (Arf) Family

The ADP ribosylation factor (Arf) family of small G proteins plays a vital role in post-Golgi sorting and trafficking, as well as maintaining the structure of the Golgi apparatus as reviewed in [8,9]. At the TGN, Arf proteins regulate the membrane recruitment of various cytosolic cargo adaptors, such as the adaptor protein (AP) complexes, and lipid-modifying enzymes, such as phosphatidylinositol (PtdIns) kinases, to initiate membrane trafficking [8,9]. The activity of Arf proteins is regulated between their membrane-associated, GTP-bound active state, and the cytosolic, GDP-bound inactive state [8,9,21] (Figure 3). The GTP and GDP states of Arfs are mediated by the Arf family guanine nucleotide-exchange factors (GEFs) and guanine nucleotide-activating proteins (GAP), respectively [8,9,22] (Figure 2A). All Arfs are recruited to membranes through an N-terminal amphipathic helix from a hydrophobic pocket [8,9]. Upon the exchange of GDP for GTP, Arfs undergo conformational rearrangement in switch 1 and 2 and the interswitch regions [21] (Figure 3B). This conformational change leads to the displacement of the myristoylated N-terminal amphipathic helix from the hydrophobic pocket [21] (Figure 3B). The exposed myristoylated amphipathic N-terminal helix then inserts into the membrane [21,23] (Figure 3B). These features distinguish Arfs from other Ras superfamily small G proteins (Ras, Rho, Rab, and Ran), which harbor a long hypervariable C-terminal domain for membrane targeting and anchoring [9,23,24].

In humans, there are five Arf small G proteins and more than 20 Arf-like (Arl) small G proteins within the Arf family [9,25], and several of them are localized at the TGN [8,9]. All five Arfs are ubiquitously expressed and can be further divided into Class I (Arf1 and Arf3, Arf2 is not found in humans), Class II (Arf4 and Arf5), and Class III (Arf6) based on sequence homology [8,9]. The Class I Arfs, which share 97% sequence identity, are the most abundantly expressed Arf small G proteins [26,27,28]. The Class II Arfs, which are 90% identical, share 81% sequence identity with Class I Arfs [29]. Unlike the Class I and II Arfs, Arf6 is the only Class III Arf and it shares the least sequence identity (66%–70%) with other Arf family members [8,9]. Arf6 does not associate with the Golgi and is the only Arf that is localized at the PM, where it regulates events such as endocytosis [8,9,28]. On the other hand, the Class I and II Arfs are localized to the Golgi apparatus, including the TGN [26,27,28,29]. The individual knockdowns of Class I and Class II Arfs showed no obvious impact on Golgi morphology, suggesting functional redundancy in some aspects of membrane trafficking [29]. However, the double knockdowns of Class I and II Arfs, in every combination, revealed insights into specific defects in secretory and endocytic trafficking [29]. Along the secretory trafficking pathway, the depletion of Arf1 + Arf3 dramatically altered the morphology of ER-Golgi intermediate compartment (ERGIC) and accumulated vesicular stomatitis virus-G (VSVG) in large ERGIC-positive puncta, indicating a block in trafficking to the cis-Golgi [29]. On the other hand, Arf1 and Arf4 depletion caused tubulation and vesiculation of the Golgi, and blocked VSVG trafficking from the ER [29]. However, not all export pathways were blocked upon the depletion of Arf1 and Arf4. For example, the trafficking of the amyloid precursor protein (APP) from the Golgi was not affected by the silencing of Arf1 and Arf4 [30]. These studies reveal that different combinations of Arfs are likely to regulate distinct trafficking pathways.

The Class I and II Arfs are also likely to regulate post-Golgi trafficking as they are also localized to the TGN [26,27,28,31]. The cargo adaptor protein complex 1 (AP-1) is dissociated from the TGN only by the depletion of both Arf1 and Arf4, however, not by the single depletion of either Arf1 or Arf4 [32]. Given that AP-1 is required for post-Golgi trafficking of several cargoes (discussed below), Arf1 and Arf4 may either function collaboratively or distinctly to regulate post-Golgi trafficking through the recruitment of AP-1. Interestingly, the specific recruitment of GDP-bound, but not GTP-bound, Arf4 and Arf5, by calcium-dependent activator protein for secretion 1 (CAPS1) to the TGN has been implicated in the trafficking of dense-core vesicles (DCV) from the Golgi network in rat PC12 cells [31]. Arf4 and Arf5 are likely to also regulate post-Golgi trafficking in neurons through CAPS, which has been shown to regulate DCV secretion of a subset of neurotransmitters in the Purkinje neurons [31,33]. The depletion of either CAPS1 or Arf4/Arf5 leads to the accumulation of a DCV marker, chromogranin, in the TGN and reduced DCV secretion [31]. In addition to the combined functions of Class I and II Arfs at the Golgi, the GTP-bound Arfs may also regulate post-Golgi trafficking of specific cargoes via the recruitment of adaptor proteins. The GTP-bound Arf1 and Arf4 have been shown to regulate the transport of mannose 6-phosphate receptors (MPRs) [34,35] and ciliary cargoes [36] from the TGN, respectively.

In addition to the recruitment of adaptor proteins to the membranes, Arfs also induce conformational changes in adaptor proteins to promote their binding to sorting signals on cargoes [37]. Structural analysis reveals that cytosolic AP-1 exists as a closed conformation and this closed conformation is unable to bind to cargo sorting signals [37] (Figure 2A). Upon the recruitment of AP-1 to the membrane, Arf1 drives the conformational opening of the core of AP-1 such that the cargo binding sites are now exposed for cargo binding [37] (Figure 2A). There are three Arf1 binding sites on AP-1 that are involved in membrane recruitment and activation of cargo binding [37] (Figure 2A). Two of the Arf1 binding sites, which involve the switch region and are required for AP-1 membrane association, are located on the N-terminus of each large subunit (β1 and γ) of AP-1 [37] (Figure 2A). Simultaneously, a third proposed binding site, which is between the back side of Arf1 and the central part of AP-1γ subunit trunk domain, leads to the assembly of 2:2 dimer of AP-1 and Arf1, and the full activation of AP-1 for cargo binding [37].

## 3. Roles of Phospholipids in Regulating Protein Sorting

Phosphoinositides (PIs) play a central role in maintaining the identity of the Golgi membranes, which is necessary for proper membrane trafficking and sorting. Each specific PI, phosphatidylinositol mono-, bis-, and trisphosphate, is generated via reversible phosphorylation at positions three, four, and five hydroxyl groups on the inositol ring of the PI [38,39]. This process is driven by distinct pool of PI kinases (PIKs) and PI phosphatases that are localized on membrane organelles [38,39]. Phosphatidylinositol 4-phosphate (PI4P) is predominantly enriched in TGN membranes, and together with Arf small G proteins, are required for mediating the recruitment of the cargo adaptors, including AP-1 and GGAs [40,41]. The distribution and levels of PI4P are regulated by PI4 kinases (PI4Ks) and phosphatases that are present at the TGN [3,40]. There are four PI4Ks: PI4KIIα, PI4KIIβ, PI4KIIIα, and PI4KIIIβ in mammalian cells, with PI4KIIα and PI4KIIIβ representing the bulk of the PI4P production at the Golgi [38,39]. The knockdown of PI4KIIα reduces PI4P levels at the TGN and causes the dissociation of AP-1 and GGAs into the cytosol, thus blocking membrane trafficking [40,41]. On the other hand, PI4KIIIβ is required for maintaining the structural integrity of the Golgi since an expression of a dominant negative PI4KIIIβ dramatically altered the Golgi organization [42]. The recruitment and membrane association of PI4KIIIβ to the Golgi is regulated by GTP-bound Arf1 [42] and another Arf1 binding partner, neuronal calcium sensor 1 (NCS1) [43,44]. Moreover, the interaction of Arf1 and NCS1 controls PI4KIIIβ activity and regulates both constitutive and regulated secretion [43,44]. Since the stimulation of PI4P production leads to the membrane recruitment of cargo adaptors, this PI pathway facilitates the coordinated sorting and trafficking of cargoes at the TGN.

The PI4P phosphatase, Sac1, has also been shown to regulate PI4P at the Golgi and is required for growth-dependent secretion in primary human fibroblasts [45]. In quiescent cells, Sac1 accumulates at the Golgi and eliminates PI4P, thereby downregulating anterograde trafficking [45]. In contrast, in human fibroblasts grown in the presence of high serum, Sac1 relocates to the ER which results in increased levels of PI4P at the Golgi, hence promoting anterograde trafficking [45].

## 4. Cargo Adaptors at the TGN

Cargo adaptors include the complexes of the adaptor protein (AP) family, as well as the highly-conserved monomeric adaptors, Golgi-localized γ ear-containing Arf-binding proteins (GGAs), which collectively play a central role in the sorting of proteins at the TGN [3,46] (Figure 1B,C). AP complexes are the most well-characterized cargo adaptors and several of these are located at the TGN. Currently, there is a family of five homologous AP complexes identified in higher eukaryotes (Figure 4) which are: AP-1, AP-3, and AP-4 that are associated with the TGN; AP-2 that regulates endocytosis at the PM; and AP-5, the most recently identified member of the AP complex family, that facilitates protein retrieval, at the late endosomes/lysosomes, back to the Golgi [18,19,47]. The AP complexes are made up of two large subunits (one of β1-5 and one of either α, γ, δ, ε, ζ; ~100 kDa), one medium subunit (μ1-5, ~50 kDa), and one small subunit (σ1-5, ~20 kDa) [18,19,47] (Figure 4). The N-terminal trunk domains of the two large subunits, the medium subunit, and the small subunit form the large globular core of the AP complex [13,20,37,48,49]. The C-terminus of each large subunit forms a long-flexible hinge region and a globular ear (appendage) domain that extend as a long projection from the core [13,20,48,49] (Figure 4). The core of the AP complex is important for membrane recruitment via binding to Arf small G proteins and phospholipids, as well as sorting signal recognition in the cargo proteins [12,13,18,19] (Figure 2A,B). On the other hand, the hinge region recruits coat proteins (i.e., clathrin), and the ear domain recruits accessory proteins which are important for membrane curvature and vesicle formation [12,13,18,19] (Figure 2A,B).

The highly-conserved GGA proteins function as ARF-dependent, monomeric clathrin adaptors to facilitate cargo sorting at the TGN. GGAs have homologous domains to the large subunits of AP complex [35,50,51,52,53] (Figure 4). In mammalian cells, there are three ubiquitously expressed GGAs (GGA1–3) that are localized at the TGN [50,51,52]. All GGAs have four domains which are comprised of the VHS (Vps27, Hrs, and STAM) domain, GAT (GGA and TOM1) domain, a flexible hinge-like region, and a C-terminal ear domain that is homologous to the AP-1 γ1-ear domain [50,51,52]. The VHS domain of GGAs has been shown to bind acidic-cluster dileucine sorting signals of cargoes, such as MPRs and sortilin [34,54]. Moreover, the VHS domain of each GGA has different binding preferences [34]. GGAs are recruited to the TGN membranes through binding of its GAT domain to membrane-associated GTP-Arf1 [55]. Similar to the AP complexes, the hinge-like domain of GGAs is required for clathrin recruitment [35,53], and the ear domain has been proposed to recruit accessory proteins [46,52].

### 4.1. Adaptor Protein (AP) Complexes at the TGN

AP-1, AP-3, and AP-4 are localized at the TGN and are predicted to generate distinct transport vesicles [18,19,20]. AP-1 is associated with clathrin-coated vesicles (CCVs) [53] and binds clathrin via the clathrin-binding motif which is located at the flexible hinge of β1 subunit [56,57]. AP-3 β3 subunit also contains a clathrin-binding motif, but AP-3 subunits are not enriched in CCVs prepared from brain or the liver [53,58]. Therefore, it remains unclear whether AP-3 is functionally capable of recruiting clathrin. On the other hand, AP-4 lacks the clathrin-binding motif and is not associated with CCVs [59,60]. Mass spectrometric protein quantification of HeLa cell lysates revealed that the number of AP-1 (~4500) and AP-3 (~4500) vesicles per cell were 30 times more abundant than AP-4 vesicles (~150) [18]. The differences in abundance suggest that AP-1 and AP-3 likely represent the major cargo trafficking pathways and the AP-4 pathway is relatively minor. Nonetheless, given that the loss of AP-4 produced drastic clinical phenotypes in humans, AP-4 may represent a specialized TGN trafficking route for a specific set of proteins that collectively represent a minor fraction of all the membrane cargoes which exit the TGN [18].


**AP-1:**


The best characterized TGN-localized AP complex is AP-1. There are two isoforms of AP-1 that are μ1 subunit, μ1A, and μ1B, which give rise to AP-1A and AP-1B, respectively [61,62]. Whilst AP-1A is expressed ubiquitously, AP-1B is only expressed in a subset of polarized epithelial cells in mammals [61,62]. A classical AP-1 mediated trafficking pathway is the delivery of newly synthesized lysosomal hydrolases bound to the MPRs from the TGN to the endosomes [63,64]. The loss of AP-1 leads to the secretion of lysosomal hydrolases instead of delivery to the lysosomes [64]. Interestingly, the loss of AP-1 also results in the accumulation of MPRs at the endosomes [64]. In order to determine the role of AP-1 in the endosomal trafficking of MPRs, a knocksideways technique to reroute the TGN-localized AP-1 to the mitochondria was developed [65]. CCVs isolated from the AP-1 knocksideways cells revealed that the MPRs (CD-MPR and CI-MPR) were significantly more depleted than the lysosomal hydrolases [65]. This finding suggests that AP-1 is also required for the trafficking of ligand-free MPRs, most likely in the retrograde pathway, back to the TGN for another round of binding and delivery of newly synthesized lysosomal hydrolases [65]. Collectively, these data support a role for AP-1 in regulating bidirectional trafficking between the TGN and endosomes.

AP-1 is also implicated with the sorting of cargoes into polarized epithelial cells and neurons [12,62,66,67]. The roles of AP-1A and AP-1B in epithelial basolateral sorting have been investigated extensively [62,67,68]. Early studies indicated that AP-1A and AP-1B differed in their localization. AP-1A is localized at the TGN and AP-1B is localized at the recycling endosomes. It was proposed that AP-1A regulates biosynthetic sorting at the TGN and AP-1B is involved in recycling of basolateral proteins from the recycling endosomes [69,70]. However, a recent study, using high-resolution microscopy, has challenged these earlier findings and demonstrated that AP-1A and AP-1B colocalize to a similar extent at the TGN and recycling endosomes in polarized Madin-Darby Canine Kidney (MDCK) epithelial cells [62]. In addition, double knockdown of AP-1A and AP-1B, but not AP-1A or AP-1B single knockdown, dramatically reduced the delivery of the newly synthesized transferrin receptor (TfR) and low-density lipoprotein receptor (LDLR) from the TGN to the basolateral PM [67]. These results suggest that both AP-1A and AP-1B compensate for each other in regulating the exit of cargoes from the TGN. However, AP-1A is likely to regulate biosynthetic trafficking to the PM independent of the recycling endosomes, since the knockdown of AP-1A, but not AP-1B, increased the trafficking of LDLR and TfR from the TGN to the recycling endosomes [67]. Interestingly, knockdown of AP-1B, but not AP-1A, significantly decreased the steady-state basolateral polarity of LDLR and TfR [67], suggesting a specific role of AP-1B in the maintenance of basolateral polarity in epithelial cells. In a follow-up study, yeast 2-hybrid and pull-down experiments demonstrated that AP-1B binds noncanonical tyrosine-based and acidic clusters sorting motifs of LDLR more strongly than AP-1A [62]. Therefore, the differential expression of AP-1B in epithelial cells may promote the sorting and maintenance of basolateral polarity of proteins that are not efficiently regulated by AP-1A. All in all, both AP-1A and AP-1B have complementary and non-overlapping roles in sorting at the TGN and epithelial basolateral polarity.

AP-1A regulates the polarized sorting of cargoes, such as the TfR, the coxsackievirus and adenovirus receptor (CAR), and the neuron-specific receptors (metabotropic glutamate receptor 1 (mGluR1), and the N-methyl-D-aspartate (NMDA)-type ionotropic glutamate receptors NR2A and NR2B), to the somatodendritic domain in rat hippocampal neurons [12,66]. AP-1 is localized to the TGN/recycling endosomes and dendrites [66], and live-cell imaging showed that the bidirectional tubular carriers moving between soma and dendrites are decorated with AP-1 [66]. AP-1 functions by excluding the cargo proteins from the transport carriers that are destined for axons [66]. The disruption of AP-1 and cargo interaction by overexpressing a dominant negative μ1A-W408S mutant resulted in misincorporation of TfR into the axonal transport carriers at the TGN/recycling endosomes in the soma/neuronal body [66]. Moreover, the overexpression of μ1A-W408S also led to mis-sorting of mGluR1, NR2A, and NR2B from the somatodendritic domain to the axons [66]. These findings highlight the importance of AP-1 in regulating somatodendritic sorting and polarity in neurons. The role of AP-1 in embryonic development is also indispensable since the knockout of the genes encoding γ or μ1 subunit in mice causes early embryonic lethality [64,71]. In addition, mutation of σ1 subunit in humans is associated with X-linked mental retardation and several neurological disorders, further highlighting the importance of AP-1 in neurological development [72,73,74].


**AP-3:**


AP-3, similar to AP-1, also exists in two isoforms, i.e., AP-3A (β3A/μ3A) and AP-3B (β3B/μ3B), where, AP-3A is expressed ubiquitously and AP-3B is neuronal-specific [58,75,76,77]. Immunofluorescent data revealed that the majority of endogenous AP-3 is associated with punctate structures that lack Tfr and it was proposed that these structures were the late endosome/lysosomes [58]. In addition, some AP-3 is associated in the perinuclear region, which likely represents the Golgi [58]. The role of AP-3 in cargo sorting to lysosomes, subsequently, came from studies identifying mutations in β3A in patients with Hermansky-Pudlak syndrome (HPS), which is a genetic disorder with defective lysosome-related organelles [78]. The AP-3-deficient fibroblasts from these patients showed an increase in cell surface expression of lysosomal-associated membrane proteins CD63, LAMP-1, and LAMP-2, as compared with fibroblasts from a healthy individual, which showed localization at the late endosome/lysosome [78]. A null allele of another AP-3 subunit, δ, was identified in an HPS mouse model, mocha [79]. The lack of AP-3 in mocha was established as the cause of pleiotropic hematologic, hypopigmentation, and neurological phenotypes in this mouse line [79]. The human and mouse phenotypes from AP-3 deficiency have one thing in common, a defective endosomal-lysosomal sorting pathway [79]. In a later study, mocha fibroblasts were also shown to display increased cell surface expression of CD63 and LAMP1 [80], similar to fibroblasts from human patients with HPS described by [78]. Using immunoelectron microscopy, LAMP-1 and LAMP-2 were often found to coincide in endosomal-associated tubules that were decorated with AP-3 in a human liver carcinoma cell line, HepG2 cells [80]. In addition, AP-3 was found in budding vesicles that were distinct to AP-1 budding vesicles at the endosomal tubular membranes [78]. These endosomal AP-3 budding vesicles were also smaller than the AP-1 budding vesicles at the TGN [80]. These studies suggest that AP-1 and AP-3 regulate distinct trafficking routes from the endosomes. Therefore, AP-3 is likely to be responsible for cargo sorting from the endosomes to lysosomes, and lysosomal-related organelles such as melanosomes.


**AP-4:**


Unlike AP-1 and AP-3, AP-4 is not as well characterized. AP-4 is the most recently discovered AP complex associated with the TGN [59,60]. It has been reported that AP-4 shows a distinct immunofluorescence staining pattern as compared with AP-1 at the TGN [59,60], suggesting that both AP complexes are likely to regulate distinct post-Golgi sorting and trafficking. Despite its relatively low abundance compared to AP-1 and AP-3 in HeLa cells [18], AP-4 is expressed in all human tissues [59,60], thus suggesting important functions in membrane trafficking. Indeed, several independent studies have reported mutations in all four subunits of AP-4 genes encoding β4 [81,82,83,84,85], ε [81,86,87,88], μ4 [84,87,89], and σ4 [81,90] in human patients. These patients displayed severe intellectual disability and progressive spastic paraplegia, with early onset of spasticity [81,82,83,84,86,87,88,89,90]. These studies also highlight the importance of each AP-4 subunit in forming a functional AP-4 complex.

In a post-mortem analysis of an AP-4μ deficient patient, histology of the brain tissue revealed neuronal malformations, neuroaxonal degeneration, and white matter loss [89]. Given the severity of the neurological phenotype associated with the loss of AP-4, it is likely that AP-4 plays an important role in brain development and neuronal trafficking. Surprisingly, in contrast to the observed phenotypes in humans, AP-4 β4*^-/-^* mice were healthy and fertile with no obvious phenotypes detected, including in the brain [91]. A poor rotor performance was the only reported fitness phenotype, hence, suggesting defects in motor neurons [91]. Nonetheless, AP-4 β4*^-/-^* mice revealed a functional role of AP-4 in polarized sorting in neurons [91]. The sorting of α-amino-3-hydroxy-5-methyl-5-isoxazolepropionic acid (AMPA) receptor to the somatodendritic domain in neurons by AP-4 requires indirect association of AP-4 with the AMPA receptor via transmembrane AMPA receptor regulatory proteins (TARPs) [91]. In AP-4 β4*^-/-^* hippocampal neurons and Purkinje cells, the AMPA receptor and TARPs are missorted into LC3-II (a lipidated form of protein light chain 3) positive autophagosomes that accumulated in the swollen axons [91]. In addition, the δ2 subtype of glutamate receptor, which had been shown to bind AP-4 μ4 in a previous study [92], also accumulated in the axons of AP-4 β4*^-/-^* Purkinje cells [91]. Conversely, the distribution of several other somatodendritic receptors, including TfR, mGluR1, and NMDA-type glutamate receptor NR1, were not affected [91]. Given that AP-1A regulates the somatodendritic sorting of TfR and mGluR1 in rat hippocampal neurons [66], these results demonstrate the selectivity of AP-4 in the post-Golgi sorting and trafficking of cargoes that are distinct for AP-1 [66,91].

In vitro and Y2H analyses have shown that AP-4 μ4 binds to canonical tyrosine-based motifs of cargoes including CD63 [60], LAMP-1, and LAMP-2 [93]. However, the binding affinities of AP-4 μ4 to these cargoes are weak, and the depletion of AP-4 did not alter the distribution of these proteins to lysosomes [94]. Therefore, in contrast to AP-3, AP-4 is unlikely to regulate TGN to lysosomal trafficking. AP-4 μ4 has also been shown to strongly bind synthetic peptides encoding the tyrosine-based sorting motif in LDLR, and the basolateral sorting motif in CD-MPR by surface plasmon resonance binding assays [95]. However, the depletion of AP-4 μ4 in MDCK cells only partially affected basolateral sorting of LDLR and CD-MPR [95]. Given that AP-1 has been shown to also regulate the trafficking of LDLR and MPRs as described above, AP-4 may function in parallel with AP-1. Consistent with this observation, AP-4 vesicle formation has been shown to be enhanced when CCVs components were depleted [18,96]. Moreover, the level of AP-4 vesicles also increased when CCVs formation was altered in AP-1 knocksideways studies from Robinson’s laboratory [18,65]. Therefore, AP-4 is likely to be a specialized cargo adaptor that regulates the TGN export of a selective set of cargoes and, in addition, may sort a subset of cargoes that are normally transported in CCVs.

An important role of AP-4 in regulating trafficking from the TGN to endosomes was uncovered in a study from the Bonifacino’s laboratory [97]. AP-4 μ4 was shown to bind to amyloid precursor protein (APP) directly, which represents a novel AP-cargo interaction [97]. AP-4 is required for the post-Golgi trafficking of APP to the endosomes in HeLa cells and neurons, a transport step which regulates APP processing [30,97,98].

More recently, AP-4 has been shown to mediate export of a multi-pass membrane core autophagy protein, ATG9A (autophagy-related protein 9A), from the TGN to endosomes and/or preautophagosomes, and is thought to promote early stages of autophagosome maturation [99,100,101]. Moreover, a recent study from the Robinson’s laboratory identified two additional proteins, SERINC1 and SERINC3 (serine incorporator 1 and 3), which co-traffic with ATG9A in AP-4 derived vesicles [101]. These vesicles were also found to associate with two newly identified cytosolic AP-4 accessory proteins, RUSC1 and RUSC3 (run and SH3-domain containing protein), which are thought to be required for microtubule plus end-directed trafficking [100].

### 4.2. Golgi-Localized, γ Ear-Containing, Arf-Binding Proteins (GGAs)

The function, and potential redundancy, of the highly conserved GGAs (GGA1-3) remain only partially defined. Evidence for a non-overlapping function amongst the GGAs comes from analyses of GGA null mice, which showed that the loss of GGA2, but not GGA1 or GGA3, is embryonic lethal [102,103]. GGAs are reported to interact and regulate the trafficking of cargoes such as MPRs and sortilin from the TGN to the endosomes [34,54]. Interestingly, GGAs and AP-1 have been shown to function cooperatively in the sorting of MPRs and their bound lysosomal enzymes into CCVs at the TGN [65,104]. However, only AP-1 is required for the trafficking of ligand-free MPRs from the endosomes back to the TGN [65]. These studies suggest that GGAs and AP-1 have cooperative roles in regulating anterograde trafficking from the TGN but distinct roles in endosomal sorting.

In addition to membrane trafficking from the TGN, GGAs have also been shown to regulate trafficking from other endosomal compartments. Firstly, GGA1 has recently been shown to mediate the recycling of cargoes such as BACE1 from the early endosomes to the recycling endosomes [105]. Secondly, GGA2 was reported to interact with epidermal growth factor receptor (EGFR) in endosomal structures [106]. The depletion of GGA2 leads to missorting of EGFR to the lysosomes via post-Golgi compartments, indicating that GGA2 is required for sustaining the level of EGFR during cell growth [106]. An increase in turnover of EGFR may explain the developmental arrest of GGA2 null mice. Lastly, the hinge region in GGA3 has two binding sites for ubiquitin and has been shown to play a role in targeting ubiquitinated proteins for lysosomal degradation [107]. For example, GGA3 regulates the lysosomal targeting and turnover of BACE1 as the depletion of GGA3 increased intracellular levels of BACE1 and elevated Aβ levels [108,109].

## 5. Cargo Sorting Signals

The cytosolic domain of cargoes consists of short, linear sequences of amino acids that function as sorting signals. The recognition of sorting signals by specific cargo adaptor proteins allows for selective incorporation of cargoes into transport carriers along the biosynthetic and endocytic trafficking pathways. The tyrosine-based (YXXΦ) and dileucine-based (DE)XXXL(LI) motifs (X denotes any amino acid and Φ denotes a bulky hydrophobic amino acid, i.e., isoleucine, leucine, methionine, phenylalanine or valine) are the best-characterized sorting signals (Table 1). The YXXΦ signals are recognized by the μ subunit of AP-1 through 4 while the μ subunit of AP-5 has yet to be identified to bind tyrosine-based sorting signals [110,111]. On the other hand, (DE)XXXL(LI) signals are recognized by hemicomplexes of AP-1 (γ-σ1), A-P2 (α-σ2), and AP-3 (δ-σ3) [110,111] (Table 1). There is no evidence suggesting that AP-4 or AP-5 binds dileucine-based signals. The AP complexes exhibit different binding affinities for individual (DE)XXXL(LI) signals. For example, AP-1 and AP-2, but not AP-3, bind the DDQRDLI signal of human major histocompatibility complex class II invariant chain (Ii) [112]. In contrast, AP-3, but not AP-2, binds the DERAPLI and EEKQPLL signals of LIMP-II and Tyrosinase, respectively [113,114] (Table 1). Therefore, these findings highlight the selectivity of sorting signals by AP complexes.

Using X-ray crystallographic analyses, the binding mechanisms of μ1A [126], μ2 [48,115], μ3A [116], and μ4 [60,93] to YXXΦ sorting signals have been determined (Figure 5A–D). The N-terminal domain of all the μ subunits is required for an AP complex assembly while the C-terminal domain binds the YXXΦ signals [127] (Figure 5A–D). The 16 β-strands within the C-terminal domain of the μ1–4 subunits are organized into subdomains A and B, giving rise to the immunoglobulin-like β-sandwich fold (Figure 5A–D). In μ2, the β1 and β16 strands in subdomain A binds YXXΦ where the Y and Φ residues fit into two hydrophobic pockets (Figure 5B). The crystal structure of μ1A was solved in a complex with the class 1 major histocompatibility complex (MHC-I) cytosolic tail, which lacks a Φ residue (Figure 5A). Although μ1A binds the Y residue similarly to μ2, the other regions of μ1A also establish additional interactions with the MHC-I tail. The YXXΦ signal of TGN38 binds μ3A at a site that is equivalent on μ2 (Figure 5C), however, with fewer contacts for stabilization [116]. In contrast to the μ1–3 subunits, μ4 exhibits the most distinct binding mechanism characterized to date for the YXXΦ signals (Figure 5D). Although the binding of μ4 to canonical YXXΦ signals is weak [93], it exhibits strong binding affinity for a unique subset of YXXΦ signals fitting the YXXΦE motif [97,99,128]. The YXXΦE motif is found in the cytosolic tails of the APP family members: APP (YKFFE), APLP1 (YKYLE), and APLP2 (YRFLE), as well as the autophagy-related protein 9A (ATG9A, YQRLE) [97,99,128]. Strikingly, the crystal structure of the μ4 C-terminal domain, in a complex with APP’s YKFFE cytosolic signal, revealed a unique binding site located on β4, β5, and β6 strands in the subdomain A of μ4 [97,128] (Figure 5D). This binding site, which also has hydrophobic pockets for Y and Φ residues, is located opposite to that of the YXXΦ binding sites on μ1–3 [97,128] (Figure 5A–D). Moreover, the crystal structure of μ4 also predicts the presence of a similar μ2 binding site and mutations in this site abolished the binding of the LAMP-2 canonical YXXΦ signal [93,97,128]. Thus, the subdomain A of μ4 is likely to have two binding sites, one for YXXΦ and one for YXXΦE. Collectively, these studies provided valuable insights that illustrate that the binding affinities and mechanism of cargo signals by μ1–4 subunits are distinct, therefore, highlighting the presence of cargo selectivity by AP complexes for regulating cargo transport to specific intracellular locations.

The binding of noncanonical tyrosine-based signals by the μ1 and μ4 subunits (Table 1) are essential for the polarized sorting of cargoes to the basolateral PM of epithelial cells and somatodendritic PM of neuronal cells, respectively [62,91]. In addition, μ4 is shown to bind a phenylalanine-based motif (Table 1), which is required for cargo sorting to the somatodendritic PM of neurons [92]. Another class of sorting signals, the acidic cluster-dileucine DXXLL motif, is recognized only by the GGAs via its VHS domain, described in [34,54,110] (Table 1). The acidic cluster-DXXLL motif is required for endosomal recycling of BACE1 (see later sections) [105,121,122] and trafficking of cargoes such as CD- and CI-MPRs between the TGN and endosomes [34,65]. Lastly, less information is known about acidic cluster sorting signals, but they are proposed to regulate sorting in the TGN [124] (Table 1). This motif is also often found in transmembrane proteins (i.e., furin) that are localized at the TGN in steady-state [110]. A recent study from the Robinson laboratory identified μ1 as an acidic cluster sorting machine [123]. In vitro studies from the same laboratory showed that the basic patch (Lys274, Lys298, Lys302, Arg303, and Arg304) of μ1 is required for the binding of acidic cluster signals [123]. Thus, these studies show the capability of cargo adaptor proteins to recognize a wide array of cargo sorting signals in a selective manner.

## 6. Accessory Proteins of AP Complexes and the Formation of Transport Vesicles

Membrane curvature followed by membrane fission from the cytosolic face of a continuous membrane (i.e., TGN) is required for the formation and release of transport vesicles, and thus allows vesicular trafficking to proceed [129,130]. The cargo adaptors by themselves do not drive membrane curvature or bud formation. This process is controlled by various accessory proteins [129,130]. The highly-conserved epsin N-terminal homology (ENTH) domain (~130–150 residues) is found in epsin family proteins (epsin1–3) [131,132], epsin-related protein (epsinR/enthoprotin) [129,132,133], and tepsin (tetra-epsin/epsin for AP-4) [134]. The formation of an amphipathic α-helix in the ENTH domain binds phosphoinositide (PI) and partially inserts into the membranes [17,135]. The ENTH domain of epsin1 is capable of promoting the formation of tubules from liposomes derived from total brain lipids [17]. Furthermore, the addition of the ENTH domain to liposomes containing 10% PI4,5P_2_ (a PI that is enriched at the PM) was shown to induce tubulation as well as fragmentation into small vesicles [17]. These observations are supported by a biophysical computational analysis conducted to analyze the impact of the ENTH domain membrane insertion on a continuous membrane bilayer [16]. This analysis, which takes into account the molecular features of lipids and proteins, predicted that the partial insertion of proteins into hydrophobic bilayers is sufficient to drive membrane fission and results in the formation of separate vesicles from continuous membranes [16]. These predictions were further validated using an in vitro quantitative liposome vesiculation assay, which showed that the ENTH domain causes membrane vesiculation and tubulation, and the degree of vesiculation also correlated strongly with the number of amphipathic helices [16]. The transient overexpression of epsin1 in BSC1 cells depleted of dynamin, i.e., a membrane-remodelling GTPase required for endocytic membrane fission events [136], rescued CCV formation in a transferrin uptake assay [16]. Together, these results suggest that the ENTH domain is capable of driving membrane curvature and fission.

Epsin1, the best-characterized epsin family member, associates with the PM by binding to PI4,5P2 [17]. Epsin1 also binds AP-2 via the appendage domains of α [137] and β2 [138] large subunits to stimulate a clathrin assembly and drive clathrin-mediated endocytosis at the PM [17]. In contrast, EpsinR is recruited to the TGN membranes in an Arf-dependent manner and preferentially binds PI4P [133,139]. EpsinR is associated with AP-1 by binding to the γ subunit appendage domain and is involved in CCV budding at the TGN [139]. However, unlike epsin1, epsinR does not stimulate a clathrin assembly [139]. Tepsin is the only AP-4 accessory protein identified to date [96,140]. Interestingly, while the membrane recruitment of epsin1 and epsinR are independent of AP-2 and AP-1, respectively, tepsin requires its AP complex, AP-4, for localization to the membranes [96]. Tepsin contains two C-terminal motifs that bind to the appendage domains of AP-4ε [140] and β4 [141]. The loss of AP-4 abolished the membrane recruitment of tepsin to the TGN in fibroblasts derived from human patients with loss-of-function mutations in AP-4 σ4 [90]. In line with these observations, a recent study using X-ray crystallography revealed that the ENTH domain of tepsin (tENTH) lacks the amphipathic α-helix and a PI binding pocket, which are found in the ENTH domains of both epsin1 and epsinR [134]. Thus, explaining why tepsin requires AP-4 for membrane recruitment. Phylogenetics and comparative genomic analyses suggest that tepsin diverged from epsins ~1.5 million years ago, and probably supports different biological functions [134]. Interestingly, tepsin contains a unique tepsin VHS-like domain (tVHS), which is not found in epsin1 and epsinR [134]. Structural data revealed that tVHS lacks an α-helix-8 that is found in VHS domains, suggesting that it lacks common VHS functions such as the binding of dileucine-based motifs [134]. Nonetheless, it is predicted that both tENTH and tVHS domains are likely to associate with a protein partner [134]. Further studies are required to fully understand the role of tepsin and its tENTH and tVHS domains in the generation of transport carriers.

## 7. Challenges and Future Directions

Many studies to date have analyzed cargo sorting in transfected cells with high levels of expression of individual cargoes. The impact of overexpression may influence the selectivity of recruitment into defined transport pathways, particularly at the TGN which has multiple transport routes. An important advancement is the development of systems to analyze the trafficking of endogenous cargo in both immortalized and primary cells. A recent innovative approach to analyze endogenous cargoes involves the development of a proteomic technique known as dynamic organellar maps, which provides a powerful complement to imaging-based analyses [142,143]. This technique combines subcellular fractionation with quantitative mass spectrometry to determine the abundance and distribution of individual proteins in the different organelle compartments in an unbiased manner [142,143]. The dynamic organellar maps technique also enables the spatial distribution of cargo proteins in each intracellular organelle to be quantified upon the loss of key transport machinery, either through genetic editing or in an inherited disease. This approach allows rapid assessment of the impact of depletion on a transport machinery component on the distribution of the entire cohort of cargo proteins that share a particular pathway. This technique should be particularly amenable to the study of minor transport pathways, such as the AP-4 mediated pathway, since AP-4 is present in low levels as compared with the other adaptor complexes.

A recent study by the Robinson laboratory utilized this approach to uncover the role of AP-5 in regulating late endosomes to Golgi recycling of various proteins including the CI-MPR, Golgi phosphoprotein 1 (GOLM1), and the Golgi integral membrane protein 4 (GOLIM4) [144]. Moreover, using the same approach, an important role of AP-4 in autophagy has been uncovered [100]. Therefore, the dynamic organellar maps technique holds considerable promise for the analyses, in an unbiased manner, of currently poorly characterized transport pathways and the impact of perturbation on the trafficking machinery.

Much of the work to date has analyzed trafficking pathways in cultured immortalized cells. Another important challenge is to define transport pathways in primary cells. For example, in melanocytes, AP-1 can partially rescue the requirement of AP-3 for the trafficking of tyrosinase from distinct tubular vesicular domains in endosome to melanosomes [145]. This study would not be possible in nonspecialized cells, which lack melanosomes. The dynamic organellar maps technique has been utilized to generate a quantitative comparison of the organellar organization in neurons [143]. Another powerful approach of this tool is to compare the trafficking studies conducted previously in cultured cells with primary cells. 

In addition, the dynamics of membrane trafficking needs to be better defined, in particular, the TGN where multiple transport pathways are generated from the same compartment. The development of a technique to synchronize the trafficking of a newly synthesized protein from the Golgi, known as the retention using selective hooks (RUSH) system [146], will enable direct analysis of the spatial partitioning of cargo destined for different transport pathways. Indeed, a recent study using the RUSH system has demonstrated a role for the transmembrane and luminal domains in the segregation of cargo proteins in the Golgi prior to recruitment by specific transport carriers [147].

An interesting issue is the potential roles of cargo adaptors in functions other than the selection of cargo into transport carriers. For example, AP-1 interacts with kinesins to promote retrograde transport along the axon in neurons [148,149]. Identifying additional non-traditional roles of adaptor proteins could be very important. A number of pathogens can highjack trafficking pathways by selective interactions with adaptor protein complexes, for example HIV Vpu hijacks AP-1 dependent trafficking [150]. It will be of considerable interest to assess whether the other adaptor protein complexes can also be selectively targeted by intracellular pathogens.

Finally, technical strategies to overcome the limitations of knockouts and knockdowns, where compensatory pathways and indirect effects may dominate, will be very important. The rapid inactivation of proteins, such as the knocksideways approach [65], will continue to be invaluable to reduce the time period between the loss of protein and the functional analyses. Moreover, the application of these technologies to in vivo whole organisms, will provide the capacity to directly examine the impact of inactivating adaptor complexes on physiological responses.

## Figures and Tables

**Figure 1 cells-08-00531-f001:**
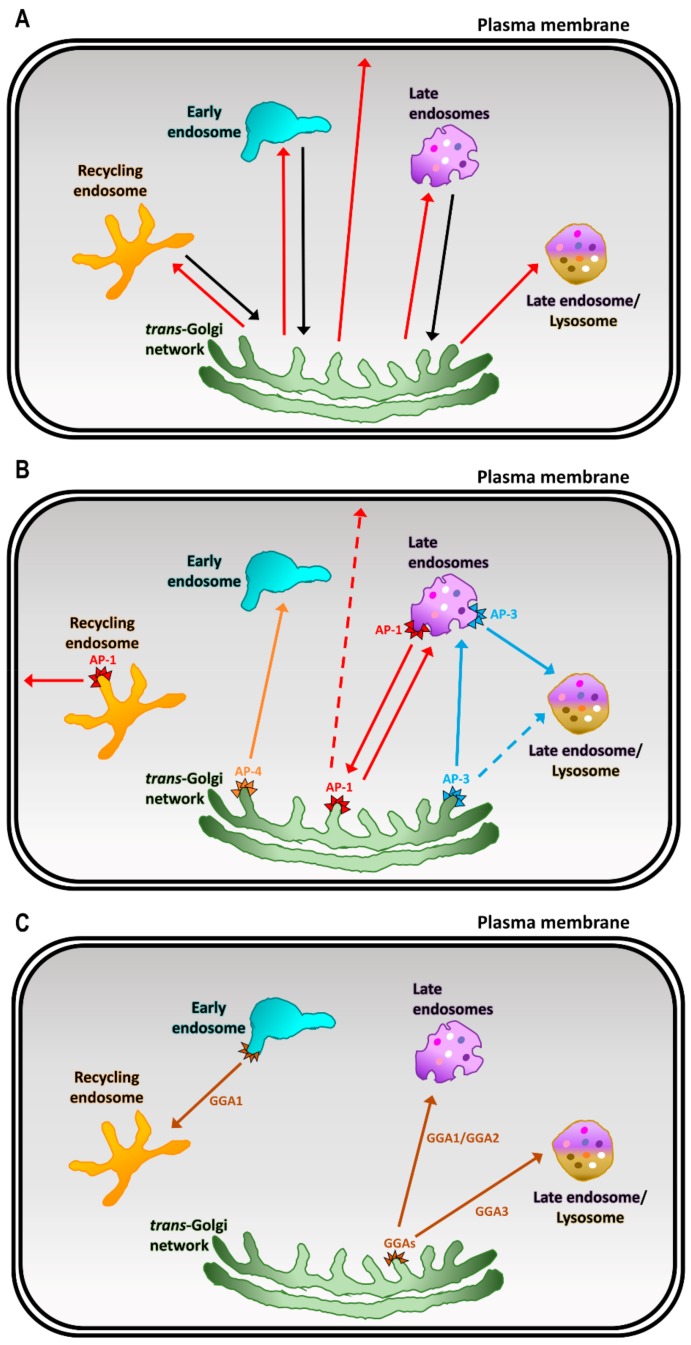
Membrane trafficking pathways at the *trans*-Golgi network (TGN): (**A**) a schematic showing the post-Golgi trafficking pathways (red arrows) and the retrograde recycling pathways (black arrows) from the plasma membrane and endosomal/lysosomal compartments back to the TGN; (**B**) the distinct trafficking pathways at the TGN are regulated by specific cargo adaptor proteins including adaptor protein (AP) complexes (AP-1 in red, AP-3 in blue, AP-4 in orange); and (**C**) monomeric Golgi-localized, γ ear-containing, Arf-binding proteins (GGA1-3 in brown).

**Figure 2 cells-08-00531-f002:**
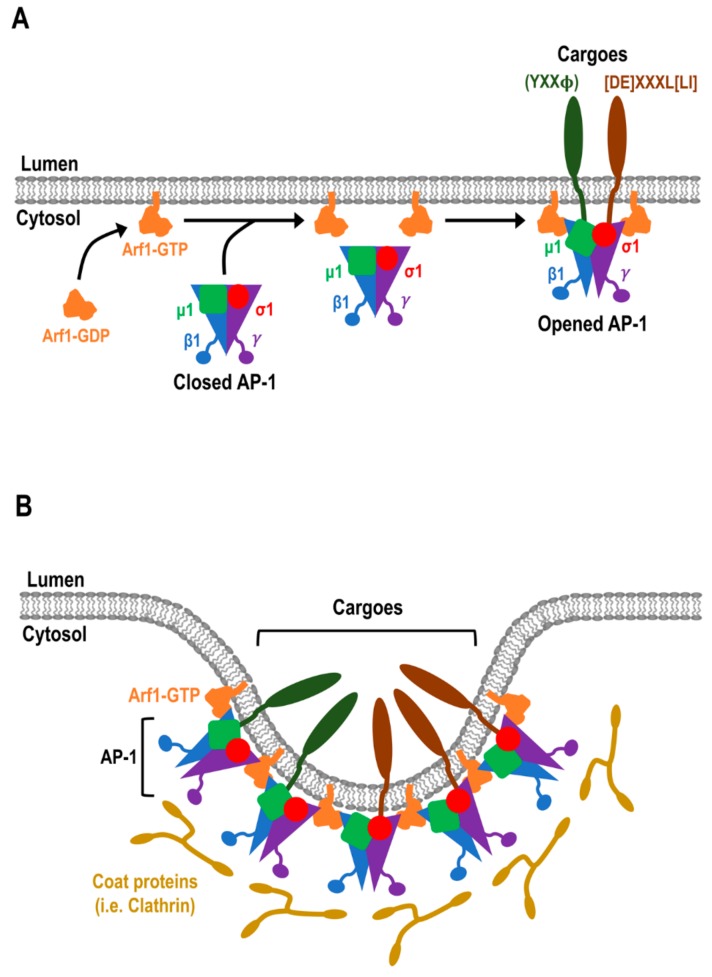
Cargo sorting at the TGN: (**A**) schematic showing the process of membrane recruitment of the adaptor protein complex, AP-1, and the binding of sorting motifs on the cytoplasmic tail of cargoes. GTP-bound membrane-associated small G protein Arf1 recruits cytosolic AP-1 to the TGN membranes. Arf1 also induces conformational change in AP-1, from closed to an open conformation, to allow the binding of cargo sorting motif(s). The μ1 subunit (green) binds tyrosine-based sorting motif (YXXΦ, green cargo) and the σ1−γ subunits bind dileucine sorting motif (DE)XXXL(LI), brown cargo). (**B**) Schematic representation of the membrane recruitment of AP complexes by active small G proteins, followed by binding of AP complex to the cytoplasmic tail of cargo proteins. Subsequently, the hinge-ear extensions of each AP complex mediate the recruitment of coat and accessory proteins to drive membrane curvature and vesicle formation.

**Figure 3 cells-08-00531-f003:**
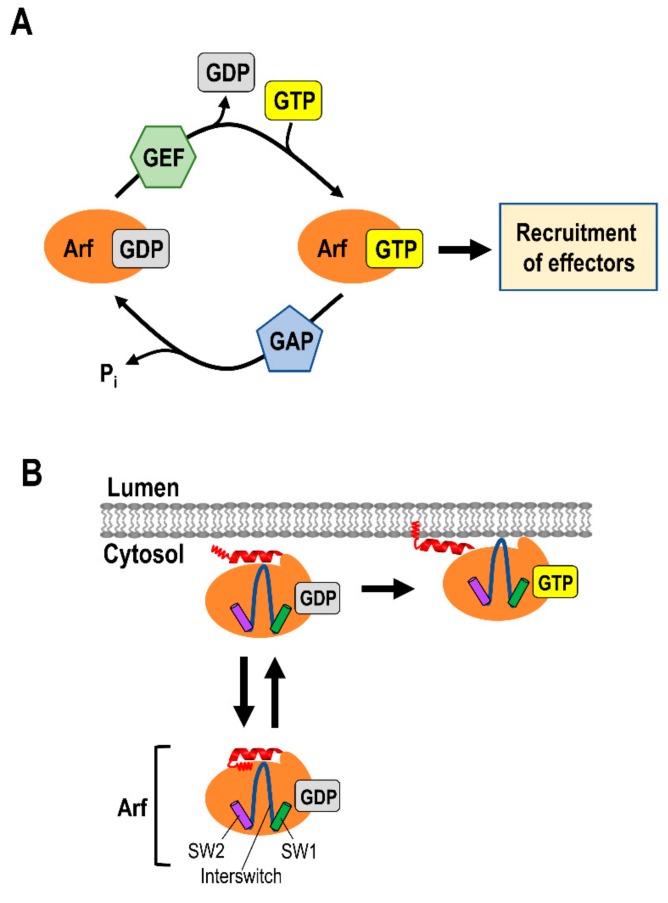
Regulation of Arf family small G proteins: (**A**) The state of the Arf small G proteins GTP binding and hydrolysis is regulated by Arf family guanine nucleotide exchange factors (GEFs) and GTPase-activating proteins (GAPs), respectively. The membrane-associated GTP-Arf then recruits effectors such as cargo adaptor proteins, membrane lipid-modifying enzymes, and additional GEFs. (**B**) GDP-Arf associates with the surface of the membrane in a reversible manner. Exchange of GDP for GTP in Arf causes a conformational change of the switch (SW1 and SW2) and interswitch regions to enter the hydrophobic pocket occupied by the myristoylated N-terminal helix (red). GTP-Arf then associates tightly with the membrane via the exposed N-terminal helix.

**Figure 4 cells-08-00531-f004:**
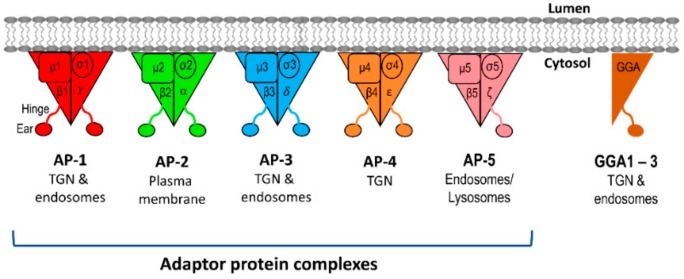
Cargo adaptor proteins: Five adaptor protein (AP) complexes, AP-1 (red), AP-2 (green), AP-3 (blue), AP-4 (orange), and AP-5 (pink), have been identified to date in higher eukaryotes. Each heterotetrameric AP complex comprises two ~100 kDa large subunits (β1–5, and either α, γ, δ, ε, or ζ), one ~50 kDa medium subunit (μ1–5), and one ~20 kDa small subunit (σ1–5). Together, they form the AP core, which is important for membrane recruitment and cargo sorting motif recognition. The C-termini of both large subunits of each AP complex give rise to the hinge and ear domains for further recruitment of accessory proteins. AP-1 is localized at the TGN/recycling endosomes and regulates bidirectional transport. AP-1 also regulates basolateral sorting in polarized cells. AP-2 is responsible for endocytosis of cargoes from the cell surface. AP-3 is localized at the TGN/early endosomes and regulates transport to the late endosomes/lysosomes. AP-4 is localized at the TGN and regulates cargo trafficking from the TGN to the early endosomes. AP-5 is localized at the late endosomes/lysosomes, and the trafficking pathway it regulates is still unclear. The three Golgi-localized γ-ear containing Arf binding isoforms, GGA1, GGA2, and GGA3 (brown), are monomeric and have similar structural protein folding to the ear domain of AP-1γ subunit. GGAs are localized at the TGN and endosomes.

**Figure 5 cells-08-00531-f005:**
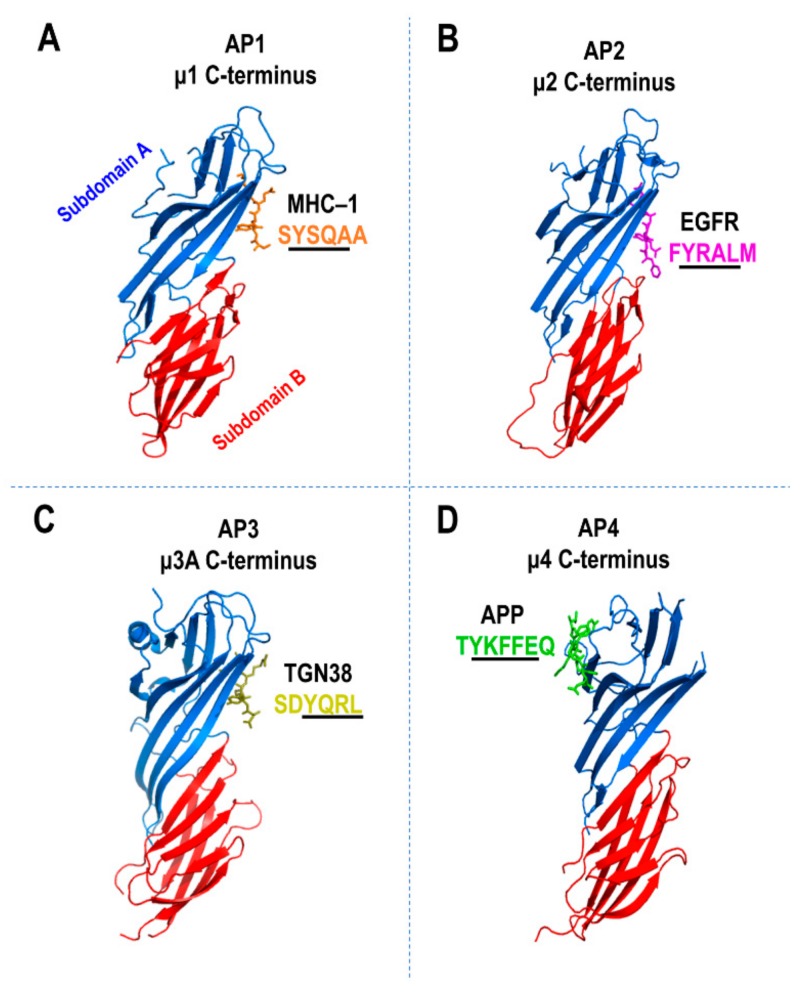
Tyrosine sorting motifs binding sites on AP complex. Crystal structures of the C-terminal domains of μ subunits of AP-1 through 4 in association with tyrosine-based sorting motifs (underlined). The C-terminal domains of μ1–4 are further subdivided into subdomain A (blue) and subdomain B (red). (**A**) Mouse AP-1 μ1 C-terminal domain in complex with class 1 major histocompatibility complex (MHC-1) peptide SYSQAA (PDB: 4EN2), (**B**) rat AP-2 μ2 C-terminal domain in complex with epidermal growth factor receptor (EGFR) peptide FYRAL (PDB: 1BW8), (**C**) rat AP-3 μ3A C-terminal domain in complex with TGN38 peptide SDYQRL (PDB: 4IKN), (**D**) human AP-4 μ4 C-terminal domain in complex with amyloid precursor protein (APP) peptide TYKFFEQ (PDB: 3L81).

**Table 1 cells-08-00531-t001:** Cargo sorting signals.

Motifs	Cargoes	Functions	Cargo Adaptor Proteins	Reference
**YXXΦ**	TGN38, TfR, Furin, MPRs, EGFR, CD63, LAMP1, LAMP2	Endocytosis, Endosomal and Lysosomal targeting	μ subunits of AP complexes	[48,79,93,110,114,115,116,117,118]
	TfR, CAR	TGN → somatodendritic domains in neurons	μ1A (AP-1A)	[66]
**[DE]XXXL[LI]**	Invariant chain	TGN → endosomes	Hemicomplexesof AP-1 γ–σ1	[112,119]
	LIMP-II, Tyrosinase	Lysosomal targeting	Hemicomplexesof AP-3 δ–σ3	[113,114]
	BACE1	Endocytosis	Hemicomplexesof AP-2 α–σ2	[120]
**YXXΦE**	APP family members, ATG9A	TGN → endosomes	μ4 (AP-4)	[97,99,119]
**Noncanonical Tyrosine-based motif**	LDLR	TGN → basolateral PM	μ1B (AP-1B)	[62]
	TARP (AMPA receptor)	TGN → somatodendritic domains in neurons	μ4 (AP-4)	[91]
**Phenylalanine-based motif**	δ2 glutamate receptor	TGN → somatodendritic domains in neurons	μ4 (AP-4)	[92]
**Acidic clusters-DXXLL**	MPRs	TGN↔endosomes	GGAs	[34]
	BACE1	Endosomal recycling	GGAs	[105,121,122]
**Acidic clusters**	Nef protein of HIV-1	Endocytosis: Downregulation of MHC-I	μ1 (AP-1)	[123]
	Furin, CD-MPR	TGN sorting		[124,125]

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
