# Peer review of "Cargo Sorting at the trans-Golgi Network for Shunting into Specific Transport Routes: Role of Arf Small G Proteins and Adaptor Complexes"

_cells, 2019, doi:10.3390/cells8060531_

Round 1
Reviewer 1 Report
I do strongly recommend this review for publication. The review is very interesting, very clear and very well written. This review well covers all the knowledges and literature needed to understand the key roles of Arfs, APs, and GGAs and their role in cargo sorting at the Trans Golgi Network.
I just have few comments:
- In the title I would be more precise and mention the role of which GTPases is discussed
- It would be interesting if authors could quickly discuss or mention the role of molecular motors and cytoskeleton in the regulation of protein sorting at the TGN.
- Is there a described role for Arf, AP and GGAs in Membrane Contact sites regulation and functions ? If yes, it would be very interesting to briefly discuss it.
- Figure 4: on the schematic please indicate clearly that there are 3 GGAs isofomrs. Either with number or with drawings.
- paragraph 2.1 lane 8 , 4500 please indicate the unit
Author Response
My responses to reviewer 1 as follows
1. In the title I would be more precise and mention the role of which GTPases is discussed
Good suggestion. We have changed the title to "..... :Role of Arf small G proteins and adaptor proteins
- It would be interesting if authors could quickly discuss or mention the role of molecular motors and cytoskeleton in the regulation of protein sorting at the TGN.
There is very little information on the role of molecular motors and the cytoskeleton on the regulation of protein sorting within the TGN. Molecular motors and the cytoskeleton certainly play a role in the transport of carriers from the TGN, however, we think this is beyond the scope of the review.
- Is there a described role for Arf, AP and GGAs in Membrane Contact sites regulation and functions ? If yes, it would be very interesting to briefly discuss it.
Membrane contact sites is an important emerging area in cell biology, as indicated by the reviewer. To the author's knowledge there is little or no information for a role for AP or GGAs in the regulation of membrane contact sites.
- Figure 4: on the schematic please indicate clearly that there are 3 GGAs isofomrs. Either with number or with drawings.
As suggested by the reviewer, we have now modified figure 4 and the legend of figure 4, to indicated there are three isoforms of GAAs, GAA1, GAA2 and GAA3. Figure 4 has been replaced with this modified Figure 4.
- paragraph 2.1 lane 8 , 4500 please indicate the unit
The unit is "number" and this has now been clarified in the text in paragraph 2.1
We thank the reviewer for these suggestions.
Paul Gleeson
Reviewer 2 Report
This manuscript reviewed the cargo sorting mediated by the individual cargo adaptor complexes. This review article focused on the key question, introduced the concepts with clear diagrams and clearly described the previous studies.
Here are a few additional minor suggestions.
1 The authors should add and introduce ubiquitin as sorting signal for endocytosis.
2 GGA proteins function as ARF-dependent, monomeric clathrin adaptors to facilitate cargo sorting and vesicle formation at TGN. It would better introduce GGA proteins have different domains which interact directly with ARF proteins, cargo and clathrin to strengthen the rationale.
3 It would better to discuss how powerful molecular, structural, genetic and imaging tools to extend what is now known.
Author Response
My responses to reviewer 2 as follows:
Here are a few additional minor suggestions.
1 The authors should add and introduce ubiquitin as sorting signal for endocytosis.
The focus of the review is cargo sorting at the trans-Golgi network. We contend that ubiquitination of cargo at the plasma membrane for endocytosis is outside the scope of this review.
2 GGA proteins function as ARF-dependent, monomeric clathrin adaptors to facilitate cargo sorting and vesicle formation at TGN. It would better introduce GGA proteins have different domains which interact directly with ARF proteins, cargo and clathrin to strengthen the rationale.
We acknowledge that the introduction to the GGA proteins on page 9 could have been better. therefore we have modified the introductory sentences on page 9, paragraph 2, along the lines suggested by the reviewer.
3 It would better to discuss how powerful molecular, structural, genetic and imaging tools to extend what is now known.
In the final section on challenges and future directions we have incorporated a discussion on the development of new molecular, genetic and imaging tools to progress the field. It is not clear to us how the reviewer would like us to extend this section. We think the balance is probably satisfactory as it stands.
We thank the reviewer for these suggestions
Paul Gleeson
Reviewer 3 Report
In Tan & Gleeson the mechanisms involved in the recognition of cargoes at the trans-Golgi network (TGN) are reviewed. The authors described the major components that regulate the sorting of cargoes into different transport carrier vesicles, including Arf proteins, adaptor proteins, GGAs and some accessory factors needed for budding and fission of transport vesicles.
The paper is clear and well written, and the mention of main issued to be elucidates and new techniques would be of interest for readers approaching to this field.
Nevertheless, there are some minor corrections that should be addressed.
One of them main one is related with Figure 3. I think this figure must be redraw. Lines connecting the different nucleotides o phosphate to the GEF and GAP respectively might induce to error, since GEF and GAP are not directly involved in the recruitment of nucleotide o the release of phosphate after GTP hydrolysis.
The reference to Arfs or Arfs-like proteins as Arf small G protein along the text is redundant once the Arf have been characterize as small at the beginning of Section1.
Page2:
line2: “I addition;
Page6:
“ Arfs are recruited to membranes through an N-terminal amphipathic helix that is myristoylated at the N-terminus”
“…displacement of the myristoylated amphipathic N-terminal amphipathic helix from the a hydrophobic pocket…”
“…Ras superfamily small G proteins including (Ras, Rho, Rab, Rac and Ran)…”
Page 7:
“Therefore, GDP- and GTP-bound forms of Arfs may have distinct trafficking regulatory mechanism” What?. Is it sure that the former sentence is the conclusion to the middle paragraph in page 7?
Page 13:
“…hydrophobic amino acid: (isoleucine…valine)…”
Page 17:
“…the impact of perturbation on trafficking machinery”
Author Response
My responses to reviewer 3 are as follows:
Nevertheless, there are some minor corrections that should be addressed.
1. One of them main one is related with Figure 3. I think this figure must be redraw. Lines connecting the different nucleotides o phosphate to the GEF and GAP respectively might induce to error, since GEF and GAP are not directly involved in the recruitment of nucleotide o the release of phosphate after GTP hydrolysis.
This is a very good point by the reviewer and we agree the presentation of Figure 3 was potentially misleading. We have re-drawn the figure and replaced Figure 3 with a new version that now indicates more accurately the role of the GEF and GAP.
2. The reference to Arfs or Arfs-like proteins as Arf small G protein along the text is redundant once the Arf have been characterize as small at the beginning of Section1.
Agree. The text on page 5 (section 1) has now deleted the reference to "small G proteins" after the initial reference in the first sentence of the first paragraph
3 Page2:
line2: “I addition;
The colon has been replaced with a comma
4. Page6:
“ Arfs are recruited to membranes through an N-terminal amphipathic helix that is myristoylated at the N-terminus”
“…displacement of the myristoylated amphipathic N-terminal amphipathic helix from the a hydrophobic pocket…”
We thank the reviewer for noting the poor phasing and have re-worded the sentence accordingly.
5. “…Ras superfamily small G proteins including (Ras, Rho, Rab, Rac and Ran)…”
We have modified the sentence on page 6 to include all the members of the superfamily, as suggested by the reviewer.
6. Page 7:
“Therefore, GDP- and GTP-bound forms of Arfs may have distinct trafficking regulatory mechanism” What?. Is it sure that the former sentence is the conclusion to the middle paragraph in page 7?
We thank the reviewer this comment. We have now deleted this sentence as it is confusing and incorrectly placed.
7. Page 13:
“…hydrophobic amino acid: (isoleucine…valine)…”
The typo has been corrected and a "(" inserted.
8. Page 17:
“…the impact of perturbation on trafficking machinery”
The phase has been corrected to " the impact of perturbation of the trafficking machinery"
We thank the reviewer for these very helpful suggestions.
Paul Gleeson